# Polydatin Protects Bovine Mammary Epithelial Cells against Zearalenone-Induced Apoptosis by Inhibiting Oxidative Responses and Endoplasmic Reticulum Stress

**DOI:** 10.3390/toxins13020121

**Published:** 2021-02-05

**Authors:** Yurong Fu, Yongcheng Jin, Anshan Shan, Jing Zhang, Hongyu Tang, Jinglin Shen, Changhai Zhou, Hao Yu, Hengtong Fang, Yun Zhao, Junxiong Wang, Yue Tian

**Affiliations:** 1Key Laboratory of Zoonosis Research, Department of Animal Science, College of Animal Sciences, Jilin University, Ministry of Education, Changchun 130062, China; fuyr20@mails.jlu.edu.cn (Y.F.); ycjin78@163.com (Y.J.); tanghy@jlu.edu.cn (H.T.); shenjinglinshen@aliyun.com (J.S.); zhouch@jlu.edu.cn (C.Z.); yu_hao@jlu.edu.cn (H.Y.); fanght@jlu.edu.cn (H.F.); zhao_yun@jlu.edu.cn (Y.Z.); jxwang19@mails.jlu.edu.cn (J.W.); letian20@mails.jlu.edu.cn (Y.T.); 2Institute of Animal Nutrition, Northeast Agricultural University, Harbin 150030, China; asshan@neau.edu.cn

**Keywords:** Apoptosis, bovine mammary epithelial cells, endoplasmic reticulum stress, polydatin, zearalenone

## Abstract

Zearalenone (ZEA) is a mycotoxin of the *Fusarium* genus that can cause endoplasmic reticulum (ER) stress and Apoptosis in bovine mammary epithelial cells (MAC-T). Polydatin (PD), a glycoside purified from *Polygonum cuspidatum*, has antioxidant properties. This study aimed to explore whether PD can alleviate ZEA-induced damage on bovine mammary epithelial cells (MAC-T). We found that incasing the concentration of ZEA (0, 7.5, 15, 30, 60, 90, 120, and 240 μM) gradually decreased the cell viability. PD treatment alone at 5, 10, and 20 μM did not affect cell viability. Follow-up studies then applied 30 μM of ZEA and 5 μM of PD to treat cells; the results showed that the ZEA + PD treatment group effectively reduced cell oxidative damage compared with the ZEA treatment group. The qPCR analysis showed that ZEA treatment significantly up-regulated the expression of ER stress-related genes, relative to the control. However, adding PD significantly down-regulated the expression of ER stress-related genes. The cell apoptosis detection results showed that, compared with the ZEA treatment group, the ZEA + PD treatment group down-regulated the *Bax* gene and up-regulated the *Bcl-2* gene expressions, which reduced the cell apoptosis rate and Caspase-3 activity. Taken together, these results indicate that PD reduces ZEA-induced apoptosis by inhibiting oxidative damage and ER stress.

## 1. Introduction

Unsuitable storage conditions and variable temperatures can promote the production of mycotoxins in harvested crops. Zearalenone (ZEA) is a fusarium toxin commonly found in feed products. Zearalenone is cytotoxic [1] to the liver, spleen [2], intestines [3], and hematopoietic cells [4]. It induces the production of reactive oxygen species (ROS) and triggers intracellular oxidative stress, which has subsequent cytotoxic and genotoxic effects [5]. Furthermore, oxidative damage reduces the performance of dairy cows and affects their disease resistance, which reduces milk yield and quality [6]. Oxidative stress is closely related to the cow’s physiology and nutritional status factors. Excessive ROS production leads to oxidative stress, loss of cellular function, and, ultimately, apoptosis or necrosis [7,8]. Our previous studies have shown that adding ZEA to bovine mammary epithelial cells promotes excessive ROS generation, endoplasmic reticulum (ER) stress, and apoptosis [9]. The balance between reduced glutathione (GSH) and oxidized glutathione (GSSH) is essential for maintaining cellular redox homeostasis. However, GSH gets depleted during protein misfolding, resulting in ROS generation [10]. Various parameters, such as malondialdehyde (MDA), reduced glutathione (GSH), total superoxide dismutase (T-SOD), and antioxidant enzyme activity, among others, are related to oxidative stress, and can indicate the cellular levels of oxidative stress [11].

ZEA was previously detected in milk samples. Therefore, whether ZEA may have similar effects on bovine mammary epithelial cells is worth studying. Studies have shown that ZEA induces mouse Leydig cell apoptosis by activating the ER stress-dependent signaling pathway [12]. The function of the ER is sensitive to the accumulation of unfolded proteins, calcium homeostasis, and redox state changes; thus, disruption of these processes can cause ER stress. Since the unfolded protein response (UPR) is an ER stress response that maintains cell homeostasis, deregulation of UPR can cause cell disorders and cell death [13]. However, previous studies found that ZEA can induce apoptosis of MAC-T cells. Mammalian cells express the UPR transducer proteins IRE1, PERK, and ATF6, which control transcriptional and translational responses to ER stress [14]. Under non-stress or physiological conditions, these proteins remain in an inactive state, and bind to the molecular chaperone BiP/GRP78, which is also the primary regulator of ER stress. Under ER stress, GRP78 decomposes from the stress sensors, thus activating them. Once activated, PERK phosphorylates the α subunit of the translation initiation factor eIF2α (eukaryotic initiation factor 2α). This activation results in an overall reduction of translation, and promotes priority translation of UPR-dependent genes (such as activated transcription factor 4 (ATF4)). The vital target of ATF4 is CHOP (C/EBP homologous protein) [15].

Oxidative stress and ROS generation are indispensable components of UPR. Therefore, ROS generation can occur both upstream and downstream of UPR. Both oxidative and ER stresses are involved in various physiological and pathophysiological conditions, and they play vital roles in cell homeostasis and apoptosis. This fact suggests that ER stress and oxidative stress have a significant correlation. For instance, our previous studies have shown that ZEA can cause ER stress and apoptosis in MAC-T cells. However, whether there are compounds that can alleviate the effects of ZEA remain unclear.

Polydatin (3,4’,5-trihydroxystibene-3-β-mono-D-glucoside; PD) is a natural precursor of resveratrol and an active ingredient in various Chinese medicines, such as *Polygonum cuspidatum*. Polydatin is the product of resveratrol and glucose, hence, it is also called resveratrol glycoside. Its pharmacological effects are similar to resveratrol, and they can be transformed into each other in the body. However, PD has a stronger antioxidant effect and stability [16]. PD may also exist in peanuts, grapes, and wine [17]. It has a wide range of positive health effects, including antioxidant [18], anti-cardiovascular diseases [19], anti-inflammatory [20], and inhibition of cancer cell growth [16], and can be used for the treatment of shock [21]. Polydatin can protect bone marrow stem cells from oxidative damage [22]; however, it is unclear whether PD can rescue cells from ER stress and apoptosis. Here, we investigated whether PD protects bovine mammary epithelial cells from ZEA-induced ER stress and apoptosis. In this study, various *in vitro* experiments were conducted to determine whether the PD-related effect caused by ZEA on MAC-T cells can be alleviated.

## 2. Results

### 2.1. The Increase in PD Decreased the MAC-T Cell Viability Due to ZEA

Cell viability was measured by the CCK-8 assay. We found that ZEA exposure significantly reduced the MAC-T cell viability, and viability decreased with increasing ZEA concentrations (Figure 1a). Cells exposed to 30 μM of ZEA had significantly lower survival than the control group (*p* < 0.001). Therefore, cells were treated with 30 μM of ZEA in subsequent experiments. MAC-T cells treated with 5–20 μM of PD had similar survival rates as the control group, suggesting that PD does not affect cell survival (Figure 1b). Given that all concentrations yielded similar results, 5 μM of PD was used for subsequent experiments. Interestingly, MAC-T cells exposed to ZEA + PD had significantly higher survival rates than cells treated with ZEA alone (*p* < 0.01; Figure 1c). The cell viability of the ZEA treatment group decreased by about 13% relative to the control group. Compared with the ZEA treatment group, the ZEA + PD treatment group increased by about 9%. The addition of PD effectively increased cell viability due to ZEA inhibition.

### 2.2. PD Reduces the ZEA-Induced Cytotoxic Effect in MAC-T Cells

The lactate dehydrogenase (LDH) activity, measured by quantifying the LDH levels released from disrupted cells into the culture media, can serve as a proxy for cytotoxicity. The results showed that LDH increased with increasing ZEA concentrations (Figure 2a). Compared with the ZEA-treated group, the LDH content of cells treated with ZEA + PD was significantly reduced (Figure 2b, *p* < 0.001). Meanwhile, the LDH content of the ZEA treatment group increased by about 43% relative to the control group. Compared with the ZEA treatment group, the ZEA + PD treatment group decreased by about 23%. These results suggest that PD lessens the cytotoxic effect in MAC-T cells treated with ZEA due to ZEA and PD interactions.

### 2.3. PD Can Inhibit ZEA-Induced Oxidative Damage

Previous results indicate that ZEA and PD can increase cell viability and reduce the LDH activity of treated cells. Therefore, whether the effect of oxidative damage is obvious requires further investigation. This study showed that the MDA level of the ZEA treatment group was significantly higher than that of the control group (*p* < 0.01, Figure 3a). In contrast, the MDA level of the ZEA + PD group was significantly lower than the ZEA group (*p* < 0.001). Compared with the control group, the MDA level of the ZEA group increased by 1.44 times. Meanwhile, the MDA level of the ZEA + PD treatment group decreased by 1.92 times relative to the ZEA group. The T-SOD content of cells exposed to ZEA was significantly lower than that of the control group (*p* < 0.01, Figure 3b). However, compared with ZEA treatment alone, the ZEA + PD treatment had significantly increased T-SOD content (*p* < 0.01). Finally, we evaluated the GSH content; we found that the GSH content of the ZEA treatment group was significantly lower than the control group (*p* < 0.001). Compared with the control group, the T-SOD activity of the ZEA treatment group decreased by 1.14 times. On the contrary, the GSH level of the ZEA + PD treatment group was significantly higher than that of the ZEA treatment group (*p* < 0.05). These results indicate that PD can inhibit the oxidative damage caused by ZEA in MAC-T cells.

### 2.4. PD Inhibits ZEA-Induced Increases in ROS Levels

The above studies indicate that ZEA causes oxidative damage to MAC-T cells, and that PD can alleviate this situation. ROS mediates oxidative stress, and is a necessary marker for detecting ROS. The standard ROS detection system uses the fluorescent probe DCFH-DA. In this study, the ZEA treatment group had significantly higher active oxygen content than the control group (*p* < 0.001; Figure 4a,b), and the ZEA + PD combination treatment significantly lowered the active oxygen content induced by the ZEA treatment (*p* < 0.001). While ZEA increased the ROS level by 2.94 times, adding PD reduced it by 1.97 times.

### 2.5. PD Inhibits ZEA-Induced ER Stress in MAC-T Cells

Both oxidative and ER stress participate in various physiological and pathophysiological conditions, and play a vital role in cell homeostasis and apoptosis. This study measured the expression levels of genes associated with ER stress to understand whether exposure to ZEA damages the ER function. The expressions of the ER stress markers *GRP78* (*p* < 0.05), *ATF4* (*p* < 0.01), *ATF6* (*p* < 0.05), *ASK1* (*p* < 0.05), and *CHOP* (*p* < 0.01) were significantly up-regulated in the ZEA treated group relative to the control group (Figure 5a–e). However, the expression levels of *GRP78* (*p* < 0.05), *ATF4* (*p* < 0.001), *ATF6* (*p* < 0.05), and *CHOP* (*p* < 0.01) in the ZEA + PD treatment group were significantly lower than the ZEA treatment group. *ASK1* expression showed a decreasing trend in the ZEA + PD group compared to the ZEA group. ZEA significantly increased the expressions of the ER stress marker genes by more than two-fold, but returned to normal levels after adding PD.

### 2.6. PD Inhibits ZEA-Induced Apoptosis

Compared to the control, treating MAC-T cells with ZEA increased the apoptotic cell rate (*p* < 0.001, Figure 6a,b); it increased from 10.22% to 29.67%. The ZEA + PD treatment group had a significantly (*p* < 0.01) lower apoptotic rate (29.67%) than the ZEA treatment group (29.06%). *Bax* expression was significantly up-regulated (*p* < 0.01), while *Bcl-2* expression was significantly down-regulated (*p* < 0.05) in the ZEA-treated group relative to the control group (Figure 6c,d). In contrast, the ZEA + PD group had significantly lower *Bax* expression (*p* < 0.001) and significantly higher *Bcl-2* expression (*p* < 0.05) than ZEA alone. Finally, we measured the Caspase-3 activity using a Caspase-3 activity assay; the Caspase-3 activity of the ZEA-treated group was significantly higher than the control group (*p* < 0.05, Figure 6e). However, the Caspase-3 activity of the ZEA + PD treatment group was significantly lower than ZEA alone (*p* < 0.01). Taken together, these results suggest that treating MAC-T cells with PD reduces the apoptotic effects of ZEA.

## 3. Discussion

Feeding cows with a moldy feed containing mycotoxins results in a lower feed intake and reduced milk production [23]. In addition, the harmful mycotoxins in feed can be secreted in milk [24]. Currently, most countries have clear guidelines and detection limits for milk toxins. Although ZEA was detected in milk, the content may not be harmful to the human body. However, for the dairy cow itself, after feeding with the feed containing ZEA, if the ZEA is transferred to the milk through the bovine mammary gland, it cannot be ignored that ZEA is harmful. Therefore, it is necessary to evaluate the toxic effects of ZEA on breast cells at the cellular level and seek ways to protect breast cells from this damage, which is beneficial for the subsequent impact on milk production.

Zearalenone, a major toxin of animal feed, deserves attention. In our previous research, MAC-T cells were used as a model to study the effects of ZEA. It was found that ZEA can inhibit cell viability, reduce mitochondrial membrane potential, and cause ER stress and apoptosis [9]. In this study, PD, a protective agent, alleviated the effect of ZEA on bovine mammary epithelial cells. We found that ZEA decreased the viability of bovine mammary epithelial cells in a dose-dependent manner, consistent with our previous findings [9]. In another study, treating human hepatoma cells (HepG2 cells) with 50 to 250 μM of ZEA for 24 h significantly reduced cell viability [25].

Recent pharmacological studies have shown that *Polygonum cuspidatum* has antibacterial, anti-inflammatory, diuretic, and other effects [26]. One of its main ingredients, PD, has various biological activities and pharmacological effects that are often cell type specific. Additionally, PD has anti-inflammatory and antioxidant properties, and can effectively treat health conditions and diseases [27]. Furthermore, PD is reported to improve arsenic damage to rat testicular cells [28] and inhibit the growth of some human tumor cells [29]. The cytotoxicity of a compound can be measured by the LDH levels released by cells [26]. Generally, an increase in LDH levels indicates increased cytotoxicity. However, adding PD significantly reduced LDH release by MAC-T cells exposed to ZEA.

SOD is an antioxidant enzyme, and its activity indirectly reflects the body’s ability to eliminate free radicals. MDA is the end product of lipid peroxidation, and the MDA levels reflect the damage caused by lipid peroxidation [16]. Measuring the levels of these compounds can provide insights into the levels of cellular oxidative damage. ZEA treatment increased the MDA content and decreased GSH and T-SOD levels, suggesting that ZEA increases oxidative damage in cells. In another study, ZEA significantly increased the Glutathione peroxidase (GPx), Catalase (CAT), and SOD activities in the testis tissue of adult Balb/c male mice, but significantly increased the MDA in the same tissue [30]. However, all of the tested concentrations of PD, including the lowest, alleviated the oxidative damage caused by ZEA on MAC-T cells. Exposing cells to both ZEA and PD reduced the MDA content and increased both GSH and T-SOD contents [28,31]. Studies have shown that PD can reduce oxidative damage in cardiomyocytes [19], consistent with the finding of the present study, where PD reduced ZEA-induced oxidative damage of MAC-T cells.

ER stress can activate mitochondrial pathways that trigger apoptosis, a reaction characterized by increased ROS production and lipid peroxidation, loss of mitochondrial transmembrane potential, activation of cysteine and acid proteases, and DNA damage [32,33]. We found that exposure to ZEA increased the ROS levels in MAC-T cells. It has previously been reported that ROS is produced in ZEA-treated leukemia cells (HL-60) [34]. However, treating cells with PD and ZEA reduced the ROS levels relative to the ZEA-treated group. Furthermore, PD can inhibit the damage induced by *S. aureus* lipoteichoic acid by attenuating ROS production [35]. These results concur with our findings that PD can effectively reduce the ROS contents of MAC-T cells.

*GRP78* and *CHOP* are markers of ER stress [36], and the up-regulation of these genes indicates increased ER stress. We found that exposing MAC-T cells to ZEA significantly up-regulated these genes, as well as *ATF4*, *ATF6*, and *ASK1*. These results are consistent with previous studies [9]. However, treating ZEA-exposed cells with PD reduced the expression levels of *GRP78*, *CHOP*, *ATF4*, and *ATF6* compared with the ZEA treatment group. These results show that PD can relieve the ZEA-induced ER stress on MAC-T cells.

The CCAAT/enhancer-binding protein homolog protein (CHOP) is a key pro-apoptotic transcription factor related to ER stress [32]. It is inadvertently known that the *Bcl-2* gene is anti-apoptotic, while the *Bax* gene is pro-apoptotic. CHOP-mediated ER stress inhibits *Bcl-2* expression, promoting *Bax* expression to induce apoptosis [37,38]. We found that cells exposed to ZEA had lower *Bcl-2* expression and elevated *Bax* expression. However, adding PD increased *Bcl-2* expression and decreased *Bax* expression. Flow cytometry evaluation of the apoptotic rate showed that ZEA treatment significantly increased apoptosis compared to the control group; however, the addition of PD significantly reduced the apoptosis rate. Although PD significantly reduced the cell apoptosis rate, the reduction degree was not large, indicating that 5 μM of PD can ameliorate cell apoptosis, but not completely inhibit it. Banjerdpongchai et al. demonstrated that ZEA could activate Caspase-3 activity in HL-60 and U937 cells in a dose-dependent manner [34]. In addition, Caspase-3 activity was significantly higher in ZEA-treated cells, and significantly decreased in the ZEA + PD treatment group compared to cells treated with ZEA. Our results are consistent with the anti-apoptotic properties of PD demonstrated by Liu et al. [39]. Taken together, our findings suggest that PD can attenuate ZEA-induced ER stress and apoptosis.

These results show that PD effectively reduces oxidative stress, ER stress, and apoptosis induced by ZEA; it may be a natural protective agent against ZEA. Therefore, this mechanism of PD should be studied further.

## 4. Conclusions

*In vitro* experiments revealed a new phenomenon that PD reduces ZEA-induced oxidative damage and ER stress in MAC-T cells by reducing ROS production, the activity of antioxidant enzymes, and the expression of ER stress-related genes. The stress subsequently alleviates cell apoptosis. These findings indicate that PD could be an effective antioxidant and potential therapeutic agent for diseases related to oxidative and ER stresses. However, the possible molecular mechanism of PD’s protective effect on ZEA-induced cell damage in MAC-T cells needs further research. Thus, *in vivo* studies are required to determine the role of PD as an effective therapeutic agent that can be used as a feed additive.

## 5. Materials and Methods

### 5.1. Chemicals and Reagents

Zearalenone (ZEA, purity >99%), hydrocortisone, penicillin-streptomycin, and insulin were purchased from Sigma-Aldrich (St. Louis, MO, USA). ZEA was dissolved in dimethyl sulfoxide (DMSO, Sigma Chemical Co., St. Louis, MO, USA) and stored at −20 °C. Polydatin (PD) was purchased from Refinsen Biotech Co., Ltd. (Chengdu, China), and dissolved in DMSO to generate a 100 mM stock solution, which was stored at −20 °C. Fetal bovine serum (FBS) was purchased from Gibco (Gaithersburg, MD, USA), while Dulbecco’s modified Eagle’s/high-glucose medium (DMEM) was purchased from Hyclone (Logan, UT, USA). The cell counting kit-8 (CCK8) was procured from Dojindo Laboratories (Kumamoto, Japan), while the FITC Annexin V apoptosis detection kit was purchased from BD Biosciences (San Jose, CA, USA). Kits to measure ROS and caspase-3 activity were obtained from Beyotime Biotechnology (Shanghai, China). Kits for detecting the T-SOD, MDA, GSH, and lactate dehydrogenase (LDH) activities were purchased from Jiancheng Bioengineering Institute (Nanjing, China). Reagents for qPCR applications included SYBR green for real-time PCR (TransGen Biotech, Beijing, China) and the RevertAid First Strand cDNA Synthesis Kit (CW0581, Beijing, China).

### 5.2. Cell Culture

The bovine mammary epithelial cell line MAC-T was kindly provided by Professor Hong Gu Lee (Konkuk University, Seoul, Korea). For *in vitro* analyses, MAC-T cells were maintained in DMEM/high-glucose media containing 10% FBS, 1% penicillin-streptomycin, 1 μg/mL of hydrocortisone, and 5 μg/mL of insulin, kept in a 37 °C incubator with 5% CO_2_.

### 5.3. Cell Viability Assay

Cell viability was measured using a CCK-8 kit (Dojindo Laboratories, Kumamoto, Japan) following the manufacturer’s instructions. Briefly, MAC-T cells were seeded in 96-well plates at a density of 1 × 10^4^ cells/well. When the cells reached 70–80% confluence, they were treated with different concentrations of ZEA (0, 7.5, 15, 30, 60, 90, 120, and 240 μM) and PD (0, 5, 10, and 20 μM) for 24 h. A cell viability test was then performed; 10 μL of CCK8 reagent were added to each well, and the cells were incubated for an additional 2.5 h at 37 °C, then randomly divided into four groups, each with three repeats. The experiments included the control, ZEA, PD, and ZEA + PD treatment groups. The absorbance at 450 nm was measured using a microplate reader (Eon, BioTek Instruments, USA), and cell viability was calculated as follows: (Treatment Group OD–Blank Group OD)/(Control Group OD–Blank Group OD).

### 5.4. Lactate Dehydrogenase Assay

We used an LDH kit (Nanjing Jiancheng Bioengineering Institute, Nanjing, China) to detect the LDH activity by the microplate method. The cells were plated in 96-well plates; ZEA and PD were treated when the cells’ confluence reached 70–80%. After 24 h, the cell culture supernatant was collected for cytotoxicity level detection. Finally, the reagents were added according to the kit instructions, and the absorbance was measured with a microplate reader at a wavelength of 450 nm.

### 5.5. Detection of GSH, T-SOD, and MDA Levels

The levels of GSH, T-SOD, and MDA were tested using the respective assay kits and following the manufacturer’s protocol. As described above, MAC-T cells were treated with varying concentrations of ZEA and PD for 24 h, and cells were collected to measure oxidation levels. The culture solution was also collected for later use. The cells were taken out with a cell scraper and transferred to a 1.5 mL centrifuge tube. Then, 500 μL of the extract was added, and the contents were mixed by homogenization. Subsequently, 100 uL of the mixture were transferred to another 1.5 mL centrifuge tube. The BCA kit determines the protein concentration. We measured the absorbance at 530 nm in the microplate reader. To determine the T-SOD content, the cells were cultivated in a similar way, then the cell protein concentration was evaluated. The kit instructions were used to add the reagents, which were mixed well and kept at room temperature for 10 min, then the wavelength of 550 nm was colorimetrically detected. To determine the GSH content, cultured cells were taken out by cell scraping and transferred to a 1.5 mL centrifuge tube. A glass homogenizer was used for mixing; 100 μL of the precipitation solution was taken and centrifuged at 3500 rpm for 10 min, and then the supernatant was taken for detection. The absorbance was measured at 405 nm in the microplate reader.

### 5.6. Measurement of ROS Production

The intracellular ROS levels of MAC-T cells were measured by a DCFH-DA kit (Shanghai, China). Cultures in multiple six-well plates were selected for the experiment. When the cell confluence reached 70–80%, they were treated and cultured in a medium containing either ZEA or PD for 24 h. Then, cultures were divided into control, ZEA, PD, and ZEA + PD treatment groups, and stained with 10 μL of DCFH-DA for 30 min at 37 °C in the dark. Cells were washed using 1× PBS to remove the unincorporated dye. The green fluorescence intensity was measured using the fluorescence microscope function of a Cytation five-cell imaging reader (BioTek Instruments, Winooski, VT, USA). Data were analyzed using the Gen5 3.03 software (BioTek Instruments, Winooski, VT, USA).

### 5.7. qPCR

The total RNA was isolated using a TRIzol reagent (Invitrogen, Carlsbad, CA, USA). Complementary (cDNA) was synthesized from 1 μg of RNA at 42 °C for 15 min, and then at 85 °C for 5 min. The SYBR Green Mix Kit was used to perform qPCR reactions. The qPCR reaction program was 94 °C for 30 s, 94 °C for 5 s, 60 °C for 15 s, and 72 °C for 10 s, respectively. A total of 40 cycles were performed. The mRNA levels were calculated by the 2^−^^∆∆Ct^ comparative method and analyzed by normalization with β-actin mRNA expression. The primers used in the qPCR analyses are listed in Table 1.

### 5.8. Flow Cytometry Detection of Apoptosis

Apoptosis was determined by staining cells with annexin V and propidium iodide (PI) using the Annexin V-FITC Apoptosis Detection Kit I (San Jose, CA, USA). MAC-T cells were first seeded in six-well plates and treated with ZEA and/or PD, as described above. After 24 h, all of the cells and culture fluids were collected for subsequent experiments. At the time of sampling, we applied trypsin without EDTA to digest the cells, and then proceeded to the cell wash step. The cells were resuspended with pre-chilled 1 * PBS (4 °C) and centrifuged at 1000 rpm for 5 min. This step was repeated thrice (trypsin was washed clean). After discarding the supernatants, cells were centrifuged, resuspended, and incubated for 15 min in 1× annexin binding buffer, 5 μL of FITC Annexin V, and 5 μL of PI working solutions. After incubation, cell apoptosis was detected by flow cytometry (Beckman-Coulter, Shanghai, China).

### 5.9. Caspase-3 Activity Assay

The caspase-3 activity assay kit (Beyotime Biotechnology, Shanghai, China) was used to measure the caspase-3 activity levels following the manufacturer’s protocols. Briefly, cells were trypsinized and centrifuged at 600× *g* for 5 min. The supernatant was discarded, and the pellet was washed with 1× PBS and centrifuged at 600× *g* for 5 min. The pellet was mixed with 100 μL of lysate reagent to resuspend cells, and the mixture was incubated for 15 min on an ice bath. After 15 min of centrifugation at 16,000× *g* and 4 °C, the supernatant was transferred to a pre-chilled centrifuge tube, and the absorbance was measured at 405 nm using a microplate reader (BioTek Instruments, Winooski, VT, USA).

### 5.10. Statistical Analyses

The data were analyzed using the SPSS statistical software package. All experiments were repeated with three independent replicates. Statistical differences among the treatment groups were calculated using one-way ANOVA, and Duncan’s Multiple Range Test was used for multiple comparisons. Data are expressed as mean ± SEM. Differences of *p* < 0.05 were considered statistically significant.

## Figures and Tables

**Figure 1 toxins-13-00121-f001:**
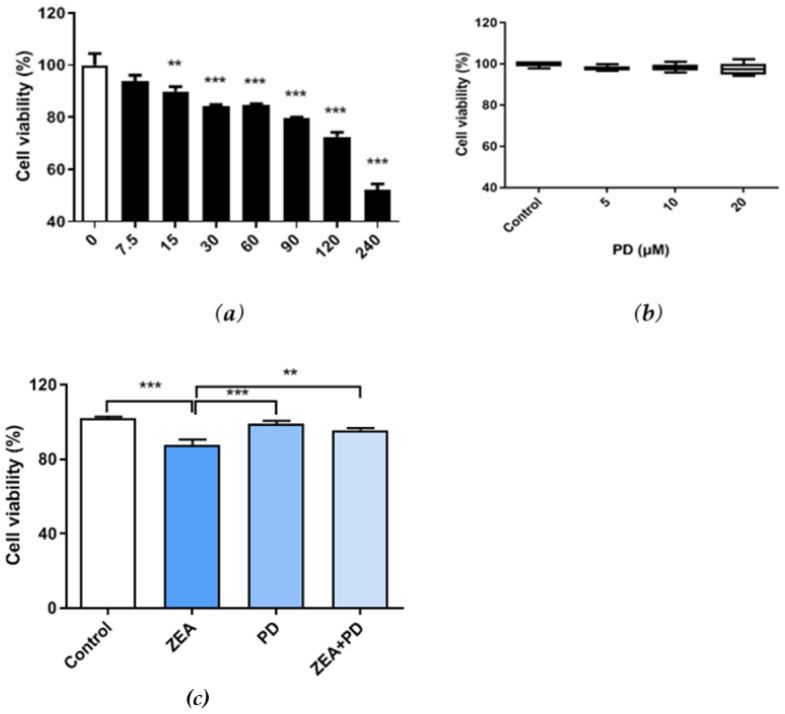
Effects of Zearalenone (ZEA) and Polydatin (PD) on the viability of bovine mammary epithelial cells (MAC-T). (**a**) Viability of MAC-T cells treated with different concentrations (0, 7.5, 15, 30, 60, 90, 120, and 240 μM) of ZEA for 24 h. (**b**) Viability of MAC-T cells treated with different concentrations (0, 5, 10, and 20 μM) of PD for 24 h. (**c**) Effects of 30 μM of ZEA and 5 μM of PD on MAC-T cell viability. Figure 1c shows the control group, ZEA treatment group, PD treatment group, and ZEA + PD treatment group. Each experiment was repeated three times. All values are expressed as mean ± SEM (*n* = 3). In the figure, ** *p* < 0.01, *** *p* < 0.001.

**Figure 2 toxins-13-00121-f002:**
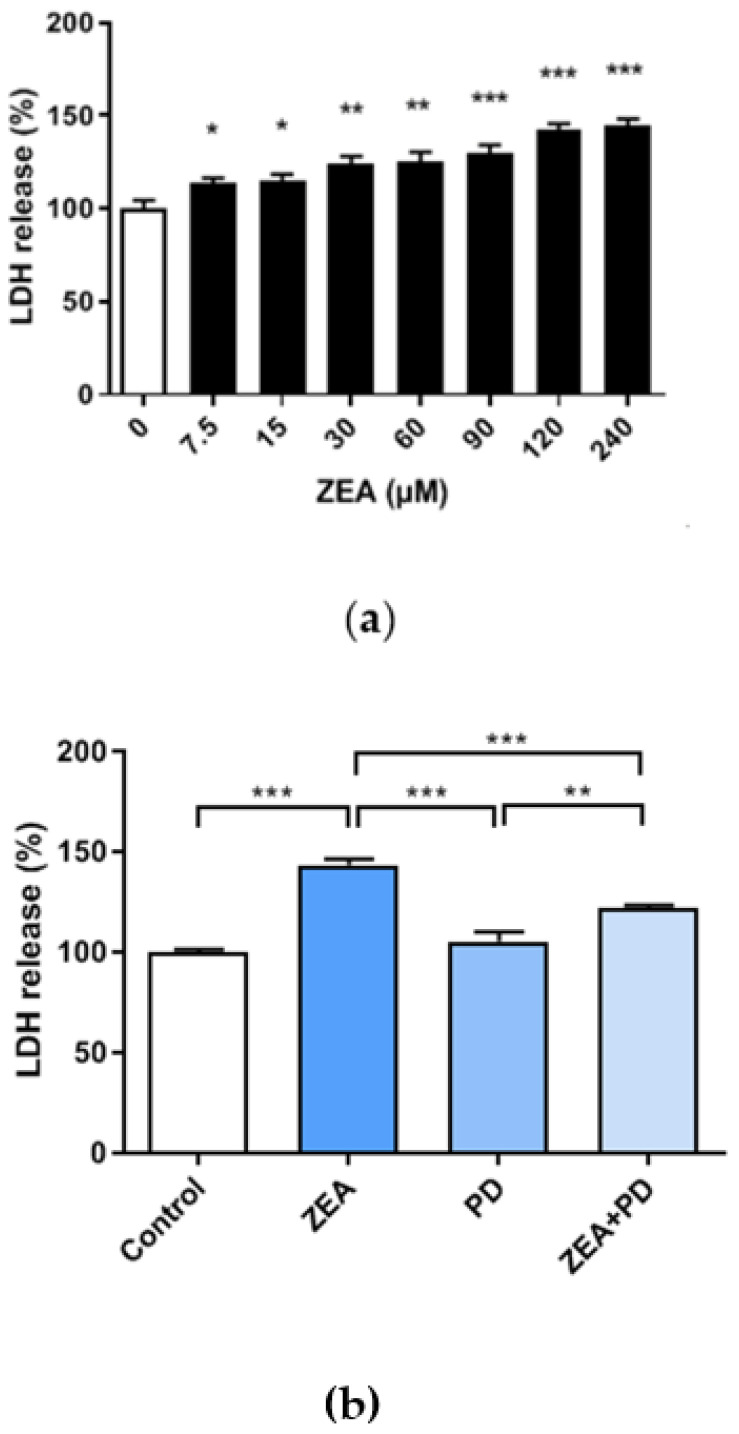
Effects of ZEA and PD on the lactose dehydrogenase (LDH) levels of MAC-T cells. (**a**) LDH released by MAC-T cells treated with varying ZEA concentrations (0, 7.5, 15, 30, 60, 90, 120, and 240 μM) for 24 h. (**b**) LDH released by MAC-T cells exposed to 30 μM of ZEA and 5 μM of PD. In the figure, * *p* < 0.05, ** *p* < 0.01, *** *p* < 0.001.

**Figure 3 toxins-13-00121-f003:**
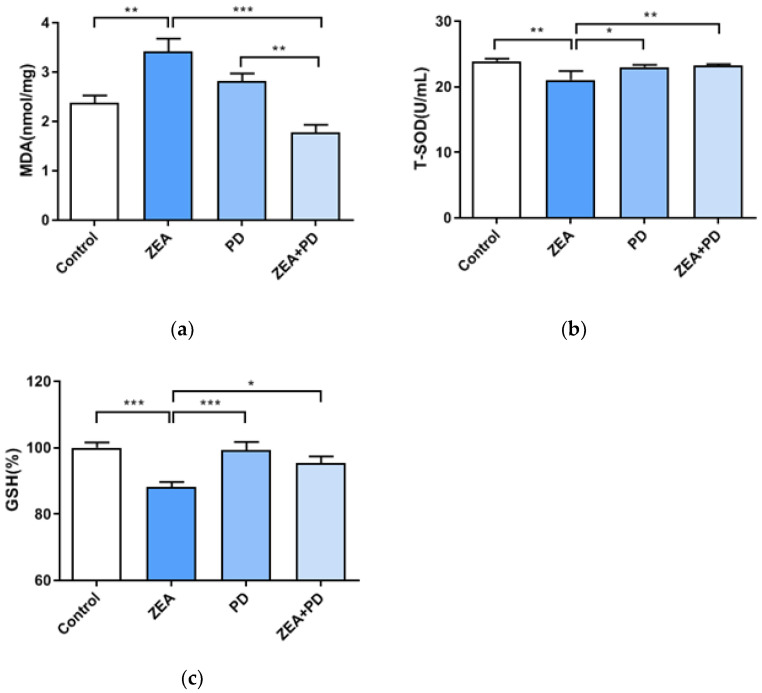
PD inhibits ZEA-induced oxidative damage in MAC-T cells. (**a**) MDA content, (**b**) T-SOD content, and (**c**) GSH content. All values are expressed as mean ± SEM (*n* = 3). In the figure, * means *p* < 0.05, ** means *p* < 0.01, and *** means *p* < 0.001.

**Figure 4 toxins-13-00121-f004:**
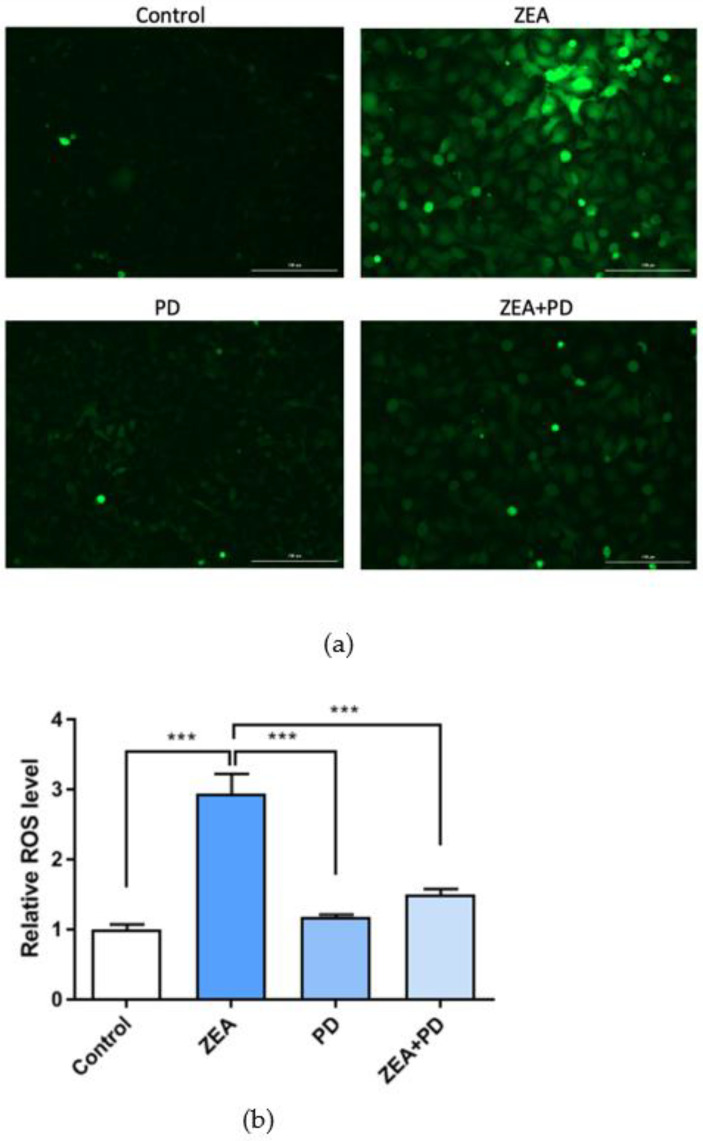
The effect of PD on ZEA-induced production of ROS in MAC-T cells. (**a**) MAC-T cells from the control, ZEA, PD, and ZEA + PD treatment groups stained with the DCFH-DA probe, and (**b**) their fluorescence intensity. In the figure, *** means *p* < 0.001. Scale bar = 200 μm.

**Figure 5 toxins-13-00121-f005:**
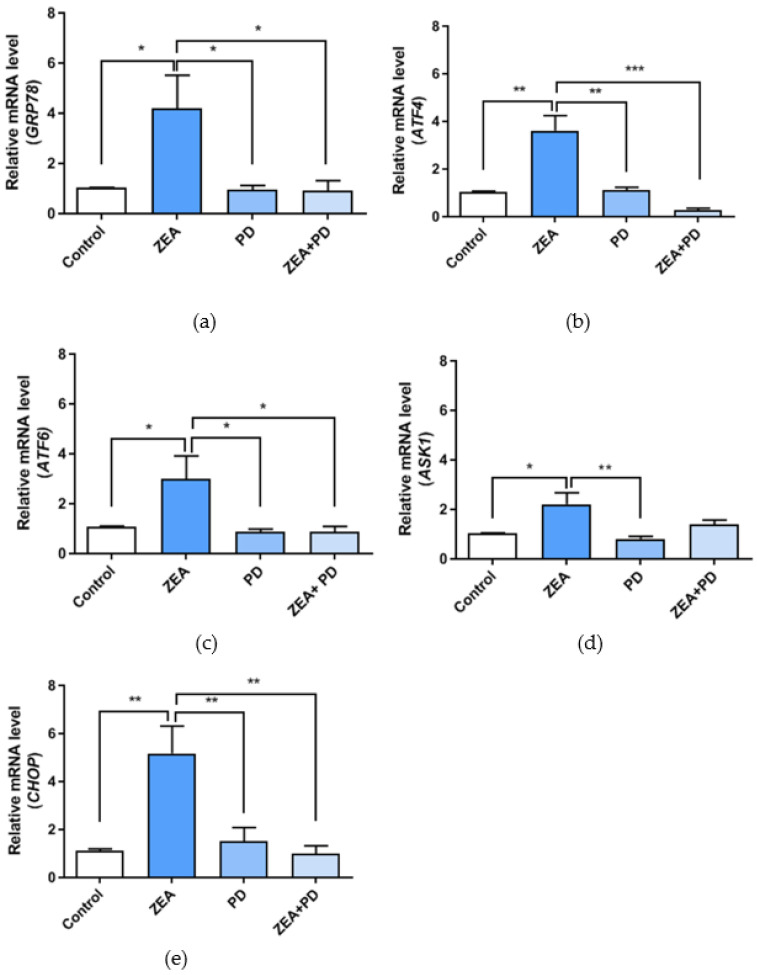
The effect of PD on the expressions of ER stress marker genes in MAC-T cells treated with by ZEA. (**a**–**e**) Analysis of gene expression levels of *GRP78*, *ATF4*, *ATF6*, *ASK1,* and *CHOP*. In the figure, * means *p* < 0.05, ** means *p* < 0.01, and *** means *p* < 0.001.

**Figure 6 toxins-13-00121-f006:**
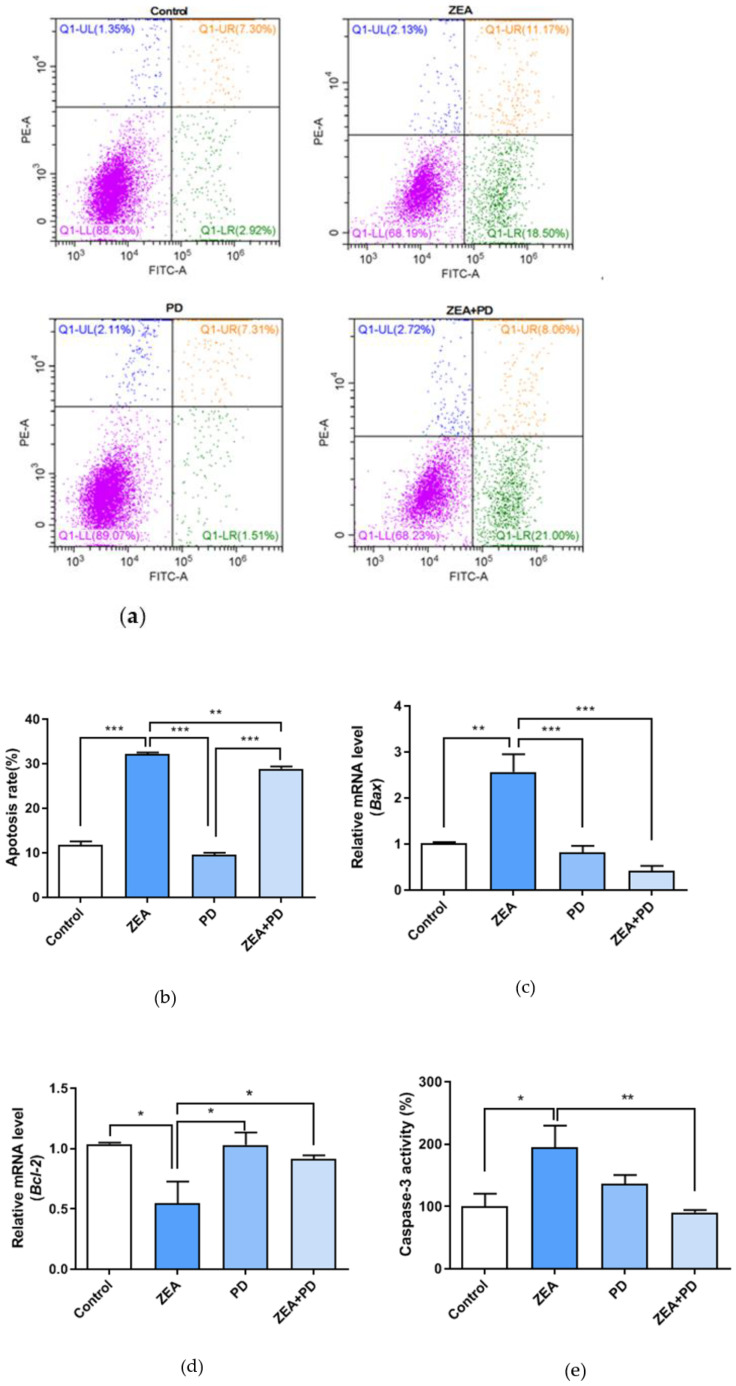
The effect of PD on ZEA-induced apoptosis in MAC-T cells. (**a**) Apoptosis measurements using annexin V/PI. The lower left panel contains annexin V and PI negative cells, while the lower right panel is annexin V positive. The upper left panel is annexin V− and PI+, while the upper right panel is annexin V+ and PI+. (**b**) Four groups of apoptotic rates. (**c**,**d**) qPCR analysis of mRNA expression levels of apoptosis-related genes *Bcl-2* and *Bax*. (**e**) Four groups of Caspase-3 activity. In the figure, * means *p* < 0.05, ** means *p* < 0.01, and *** means *p* < 0.001.

**Table 1 toxins-13-00121-t001:** Gene name and PCR primer sequences.

Gene	Forward Primer	Reverse Primer	GenBank Accession No.	Product Size (bp)
*β-actin*	5′-CCCTGGAGAAGAGCTACGAG-3′	5′-GTAGTTTCGTGAATGCCGCAG-3′	NM_173979.3	130
*GRP78*	5′-CGACCCCTGACGAAAGACAA-3′	5′-AGGTGTCAGGCGATTTTGGT-3′	NM_001075148.1	198
*ATF4*	5′-AGATGACCTGGAAACCATGC-3′	5′-AGGGGGAAGAGGTTGAAAGA-3′	NM_001034342.2	190
*ATF6*	5′-ATATTCCTCCGCCTCCCTGT-3′	5′-GTCCTTTCCACTTCGTGCCT-3′	XM_024989876.1	103
*ASK1*	5′-GCTATGGAAAGGCAGCCAGA-3′	5′-TCTGCTGACATGGACTCTGG-3′	NM_001144081.2	160
*CHOP*	5′-GAGCTGGAAGCCTGGTATGA-3′	5′-CTCCTTGTTTCCAGGGGGTG-3′	NM_001078163.1	90
*Bax*	5′-GCTCTGAGCAGATCATGAAGAC-3′	5′-CAATTCATCTCCGATGCGCT-3′	NM_173894.1	167
*Bcl-2*	5′-GATGACCGAGTACCTGAACC-3′	5′-AGAGACAGCCAGGAGAAATCA-3′	NM_001166486.1	123

## Data Availability

Data sharing not applicable.

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
