# Peer review of "Polydatin Protects Bovine Mammary Epithelial Cells against Zearalenone-Induced Apoptosis by Inhibiting Oxidative Responses and Endoplasmic Reticulum Stress"

_toxins, 2021, doi:10.3390/toxins13020121_

Round 1
Reviewer 1 Report
The techniques carried out in this study are sound and have a good rationale for their use.
Minor issues are:
Figure 1c shows that 30 micromolar (uM) ZEA was used to induce oxidative stress and decreased cell viability which was reversed by addition of PD. The concentration of ZEA used in these experiments was rather low (30uM) only inducing ~10% cell death. What was the rationale for using 30uM ZEA? In my view, it would be much better to use a ZEA concentration that induces a greater degree of cell death, such as 90uM or 120uM, to determine if PD does indeed protect against oxidative stress and cell death.
In the results section describing ER stress markers (lines 118 - 125), there should be references for the previous use of the targets as markers of ER stress.
The graph in Fig. 6B suggests that PD may not reduce apoptosis by much when applied with ZEA. Although statistically significant, would such as small reduction in apoptosis due to inclusion of PD with ZEA be 'biologically significant'? I would suggest addressing this in the Discussion section somewhere.
For all figures throughout the manuscript, the use of a, b, c, etc. are confusing as they do not indicate which groups are being compared to one another. I would suggest indicating which treatment groups/controls are being compared.
Author Response
Q:Figure 1c shows that 30 micromolar (uM) ZEA was used to induce oxidative stress and decreased cell viability which was reversed by addition of PD. The concentration of ZEA used in these experiments was rather low (30uM) only inducing ~10% cell death. What was the rationale for using 30uM ZEA? In my view, it would be much better to use a ZEA concentration that induces a greater degree of cell death, such as 90uM or 120uM, to determine if PD does indeed protect against oxidative stress and cell death.
A:Thanks for the reviewer's question. First, when 30μM ZEA was used to treat MAC-T cells, the cell viability results showed that compared with the control group, P<0.001. The significance was higher. Secondly, in actual production, ZEA can be detected in milk. According to recent literature, we know that the concentration range of ZEA detected in milk is 0.51~5.09μg/L(Flores-Flores and Gonzalez-Penas, 2018). It shows that ZEA can be transported through the blood into the milk. According to the basic theory of milk synthesis in the breast, we know that it takes about 400L of blood to produce 1L of milk. Calculated according to the highest concentration of ZEA detected in milk, 5.09μg / L, it means that the content of ZEA in 400L blood should be at least 5.09μg before it can be transferred into milk. We can then push back the actual concentration of ZEA in the blood: 5.09μg ÷ 400L = 0.017225 μg / L, the molecular weight of ZEA is 318.36g / mol, then the molar concentration of ZEA in the blood is: 0.017225μg / L ÷ 318.36 g / mol = 39.97 μM. Indicating that the 30 μM concentration selected by the cell assay is physiologically significant within the actual biological dose range. Similarly, according to the previously provided references, milk contains an average of 6.9 ppb of ZEA (El-Hoshy, 1999), which has exceeded the concentration of 5.09 μg/L in the latest reference milk. Similarly, the 30 μM concentration we selected is also less than Actual measured value. And whether it is actual in vitro research or in vivo research, as the reviewer proposed that the concentration of ZEA should be in accordance with the actual physiological concentration, therefore, by analysis, our final selection of 30μM concentration is in line with the physiological concentration dose, so it is used in cell culture for in vitro studies. The concentration added.
Flores-Flores, M. E. and E. Gonzalez-Penas. 2018. Short communication: Analysis of mycotoxins in Spanish milk. J Dairy Sci 101(1):113-117.
Q:In the results section describing ER stress markers (lines 118 - 125), there should be references for the previous use of the targets as markers of ER stress.
A:Thanks for the reviewer's question. Regarding the issue of ER stress gene selection, we will make a supplementary explanation in the preface. L50-60.” However, through previous studies, it was found that ZEA would indeed induce apoptosis of MAC-T cells. Mammalian cells express the UPR transducers IRE1, PERK, and ATF6, which control transcriptional and translational responses to ER stress[14]. Under non-stress or physiological conditions, these proteins remain in an inactive state and bind to the molecular chaperone BiP/GRP78, which is also known as the main regulator of ER stress. Under ER stress, GRP78 decomposes from the stress sensors, causing them to activate. Once activated, PERK will cause the phosphorylation of the α subunit of the translation initiation factor eIF2α (eukaryotic initiation factor 2α), which leads to the overall reduction of translation and allows the UPR-dependent genes (such as activated transcription factor 4 (ATF4)) Priority translation. The important target of ATF4 is CHOP (C/EBP homologous protein)”
Q:The graph in Fig. 6B suggests that PD may not reduce apoptosis by much when applied with ZEA. Although statistically significant, would such as small reduction in apoptosis due to inclusion of PD with ZEA be 'biologically significant'? I would suggest addressing this in the Discussion section somewhere.
A:Thanks for the reviewer's question. Regarding the apoptosis detection results of the flow cytometry results in Figure 6A and B, it is true that the apoptosis rate of the ZEA+PD co-treatment group did not decrease to a large extent, but the decrease was statistically significant. Therefore, we believe that PD plays a role in relieving and alleviating cell apoptosis caused by ZEA, but is not an effective inhibition. We will add it in the discussion section.L275-279.” Flow cytometry mediated measurement of apoptotic rate showed that ZEA treatment significantly increased apoptosis compared to the control group, however, addition of PD significantly reduced the apoptosis rate. Although PD significantly reduced the rate of cell apoptosis, the degree of reduction was not large, indicating that 5 μM PD can reduce the occurrence of cell apoptosis, but it cannot completely inhibit it.”
Q:For all figures throughout the manuscript, the use of a, b, c, etc. are confusing as they do not indicate which groups are being compared to one another. I would suggest indicating which treatment groups/controls are being compared.
A:Thanks to the reviewers for their comments, we have revised the result pictures. Use "*" to indicate.
Reviewer 2 Report
Comments and Suggestions for Authors
The manuscript (ID:toxins-1084421) “Polydatin protects bovine mammary epithelial cells upon Zearalenone-induced apoptosis through inhibiting oxidative responses and endoplasmic reticulum stress” is dedicated to the topic of importance in the field of agriculture and bovine cattle. It looks at the problem of mycotoxin (Zearalenone -ZEA) in crops, its effect on bovine epithelial cells and possible compound – polydatin (PD) that could be considered as antagonist for this mycotoxin ZEA.
While I was able to understand the manuscript, one must admit that manuscript is poorly written. Several parts lack more information. Also, the consistency problem is throughout the whole manuscript (e.g. graphs and p values, description of centrifugation speed must be either “rpms” or “x g”, but, please, do not mix).
It is hard to draw the conclusions of self-citation, but I noticed that there could be some (which is in line with allowed % of self-citation).
Experimental setup also lacks extra information (see the questions below).
Authors MUST read the principles of writing the scientific paper and must give the manuscript to some native English speaker. Structures of sentences often resemble the essays that students write for some assessments.
Major revisions (R) and questions (Q) (note R can include Q)
Q1: The cells were treated with ZEA and PD simultaneously showing some level of antagonism between two compounds. What was the overall idea on application of PD in real-life situation? Would you suggest: a) giving PD as additive to the feed or b) you are planning to use PD as treatment for such animals (that have eaten the contaminated food)? This information is absent in the manuscript and makes a big difference in experimental setup. If the answer would be b) then the experimental setup is incorrect and asks for primary treatment of the cells with ZEA and addition of PD only later.
Q2: It has been known that DMSO can alone be cytotoxic and can alter gene expression. If the ZEA and PD was diluted in DMSO, why you did not include the DMSO control in the experiments?
Q3: In quantitative PCR there are guidelines MIQE (Stephen AB, Vladimir B, Jeremy AG, Jan H, Jim H, Michael K, et al. The MIQE Guidelines: Minimum information for publication of Quantitative Real time PCR Experiments. Clinical Chemistry. 2009. 55:4.) that propose usage of MORE than only one reference gene! What was the rationale to use only one reference gene – beta-actin in your study?
R1: The introduction lacks scientific soundness and important information e.g., what type of molecule is PD. Rationale must be improved (see question 1).
R2: Authors write (line 33-35): “Zearalenone induces apoptosis in 33 mouse Leydig cells by activating the endoplasmic reticulum (ER) stress-dependent signaling pathway [6].” However, I miss the connection among Laydig cells (that are in seminiferous tube and I important in spermatogenesis), mammary epithelial cells and milk yield and quality. Probably there is a connection, but authors must include this explanation in the introduction.
R3: please, give rationale/explanation about usage of mammary epithelial cells and no other cells!
R4: Introduction needs more detailed information on relationship between oxidative stress and endoplasmic reticulum (ET) stress.
R5: Each result undersection must be re-written giving exact numbers (where it is important) not only mentioning what is increased or decreased. Also start each section with short introduction on why authors look at certain parameters.
R6: Choose only ONE TYPE of statistical significance. Usually, scientific graphs have the * symbol for p values.
R7: Line 72-73 “In Fig. 1C, they are the control group, 72 the ZEA treatment group, the PD treatment group, and the ZEA+PD treatment group.” – this statement belongs to materials and methods NOT in results.
R8: Line 78 “LDH activity can serve as a proxy for cytotoxicity” – when mentioning for the first time, please, state the full name of LDH.
R9: Line 81-83 “Compared with the ZEA-treated group, the LDH content in the cells treated with ZEA + PD was significantly reduced (Figure 2B, P<0.01).” – HOW much! Need numbers!
“PD reduces the LDH of MAC-T cells because ZEA increases.” – is this a conclusion? Why it is here. Also the sentence lacks scientific style. E.g. the results can lead to suggestion that in …cells treated with ZEA, PD lessens the cytotoxic effect due to interactions of ZEA and PD. -this is only an example.
R10: Section 2.3. probable typo in the title, Cxidative stress – needs to be Oxydative stress. Also, what does MDA, T-SOD and GSH stands for and why they are important (probably it is better to mention it in introduction)?
R11: In method about measuring the levels of ROS, authors state that intensity was measured “using the fluorescence microscope function of a Cytation 5 cell imaging reader”. Does it include counting the cells? How many view fields were counted? Is it really quantitative or more semi0quantitative method? If it is quantitative methods, please provide actual (or average) numbers in each group including standard deviation!
R12: Figure 4 need more visible scale bars! Also, in the legend for figure 4: “In the figure, A and B means P<0.01” – how can two signs mean the same p value?
R13: Section about ET stress and gene expression – please provide (either in introduction or in discussion or even here) the information-rationale to use those particular genes as ET stress markers. Also, mention fold changes!
R14: Figure 5 – for better visibility all graphs should have identical Y axis scale. Consistency!
R15: Primer table must be in section Materials and Methods.
R16: Section 2.6. about apoptosis: authors should start with explanation how the apoptosis was detected and then continue with findings. About Bax and Bcl2 – at least on one place mention which one was PRO and which was ANTI apoptotic protein. Please, remember that gene names must be written in italica.
R17: Figure 6 A: at least in legend, provide information what is what in those FACS dot plots? Also, each quadrant description (e.g. in control dot plot Q1-UL, Q1-UR etc is very confusing. Name them correctly. In addition, I would suggest to put pictures A and B beside each other and then all three graphs with gene expression changes and caspase activity in other line. Probably it is a fold change depicted in figure 6. C and D – then it must be mentioned in the results and also number must be given with standard deviations.
R18: Discussion beginning (line 155): If it (mycotoxin) is in the milk, how it gets there? And is it still harmful for us as milk-drinkers? Or is the problem here only about cows giving more milk?
R19: line 163 - Again, what was the rationale to choose PD? Please provide the information. What type of molecule is PD?
R20: line 176 “… to treat health-related diseases” – do we have health-non related diseases? Line 177 “…kill various human tumor cell lines” – styling is poor.
R21: Discussion has a lot of repetitious information e.g. lines 178-180 (about LDH) repeats Lines 168-170.
R22: Line 187-189 “The effect of ZEA on GPx, CAT, and SOD in the 187 testis tissue of adult Balb/c male mice was significantly decreased, while MDA in the 188 same tissue was significantly increased [27].” – How would you explain that increase?
R23: Lines 190-192 “Exposing cells to both ZEA and PD reduced MDA content and increased GSH and T-SOD content [28, 24].” – but is it done in rats! Please be precise and mention full information. Also, the next sentence states the similarities of your work with Jiang et al. [15]. Please, provide in which model they observed the statement. How it related with previous sentence?
R24: while giving some reference on GRP78 and CHOP as possible ET stress markers, nothing is said about ATF4, ATF6 and ASK1.
Overall, discussion must be re-written.
R25: Materials and Methods – Line 265: “The cell fluid was collected….” – did you mean “cell media” or “cell supernatant”?
R26: Section 4.5. Detection of GSH, T-SOD and MDA levels – this section need more explanation. I will not go and look for specific protocol (manufacturer must be mentioned in methods, not only in materials).
R27: Section 4.6. is very confusing: Line 277-279 “When the cells confluence [strange styling] 70-80 %, they were treated and cultured in a medium containing ZEA or PD for 24 h. Then they were divided into control, ZEA, PD and ZEA+PD treatment groups.” – Grouping would be logical before treatment, but how were you able to take control group, if you treated cells with ZEA or PD?
R28: Section 4.7. qPCR - How much RNA was used for cDNA synthesis? Why two RT reagents? Why repeat what was written in section 4.1.? Do not write it in 4.1. section then. Line 289 “First strand cDNA was synthesized according to manufacturer’s instructions.” - which manufacturer? In my opinion this sentence is irrelevant.
R29: please, provide the reference also for the Fold change method (delta delta Ct).
R30: Section 4.8. and 4.9. please be consistent with description of the speed of centrifugation, using either “rpm” or more widely used “… x g”.
R31: Conclusions are to general and needs to be more conclusive (for that you need a clear rationale and need to give bigger picture (see Q1)).
Author Response
The manuscript (ID:toxins-1084421) “Polydatin protects bovine mammary epithelial cells upon Zearalenone-induced apoptosis through inhibiting oxidative responses and endoplasmic reticulum stress” is dedicated to the topic of importance in the field of agriculture and bovine cattle. It looks at the problem of mycotoxin (Zearalenone -ZEA) in crops, its effect on bovine epithelial cells and possible compound – polydatin (PD) that could be considered as antagonist for this mycotoxin ZEA.
While I was able to understand the manuscript, one must admit that manuscript is poorly written. Several parts lack more information. Also, the consistency problem is throughout the whole manuscript (e.g. graphs and p values, description of centrifugation speed must be either “rpms” or “x g”, but, please, do not mix).
It is hard to draw the conclusions of self-citation, but I noticed that there could be some (which is in line with allowed % of self-citation).
Experimental setup also lacks extra information (see the questions below).
Authors MUST read the principles of writing the scientific paper and must give the manuscript to some native English speaker. Structures of sentences often resemble the essays that students write for some assessments.
Major revisions (R) and questions (Q) (note R can include Q)
Q1: The cells were treated with ZEA and PD simultaneously showing some level of antagonism between two compounds. What was the overall idea on application of PD in real-life situation? Would you suggest: a) giving PD as additive to the feed or b) you are planning to use PD as treatment for such animals (that have eaten the contaminated food)? This information is absent in the manuscript and makes a big difference in experimental setup. If the answer would be b) then the experimental setup is incorrect and asks for primary treatment of the cells with ZEA and addition of PD only later.
A:Thanks for the reviewer's question. PD is an effective ingredient of traditional Chinese medicines such as Polygonum cuspidatum, which is a product of the combination of resveratrol and glucose. Therefore, it is also called resveratrol glycoside. Its pharmacological effects are similar to resveratrol and can be transformed into each other in the body, but PD has more antioxidant effect and stability. This is the characteristic of PD. The use plan of PD is to use it as a feed additive to protect the mammary glands caused by ZEA in the feed, not for therapeutic use. Therefore, the experimental design was carried out and added at the same time to see its protective effect. First of all, we took into account the simultaneous and separate addition of ZEA and PD during the preliminary experiment. The detection of cell viability and lactate dehydrogenase showed that adding ZEA and PD at the same time had no difference in the effect of treating PD after adding ZEA. The selected PD has a lower concentration at the same level and will not cause toxic effects on cells, so it is selected to be added at the same time.
On the other hand, based on the chemical structure of ZEA and the structure of PD, there is no possibility of antagonism. Therefore, there is no PD antagonizing ZEA, reducing the concentration of ZEA and reducing the toxic effect on MAC-T cells. We consider that PD and ZEA enter the cell together, and PD hinders the signal pathway of ZEA. The specific mechanism is not very clear.
Q2: It has been known that DMSO can alone be cytotoxic and can alter gene expression. If the ZEA and PD was diluted in DMSO, why you did not include the DMSO control in the experiments?
A:Thanks for the reviewer's question. There are related literature references for choosing DMSO to dissolve ZEA and PD, and our dosage is less than one-thousandth. And we did the cell viability test of the blank control and DMSO dosage, and the results proved that the DMSO content we used was not significantly different from the blank control group, and it would not affect cell viability.
Q3: In quantitative PCR there are guidelines MIQE (Stephen AB, Vladimir B, Jeremy AG, Jan H, Jim H, Michael K, et al. The MIQE Guidelines: Minimum information for publication of Quantitative Real time PCR Experiments. Clinical Chemistry. 2009. 55:4.) that propose usage of MORE than only one reference gene! What was the rationale to use only one reference gene – beta-actin in your study?
A:There are many kinds of β-actin as the commonly used internal control. There are also GAPDH and so on. In breast cells, the expression of β-actin is relatively stable and consistent. In many breast cell articles, β-actin is used as a fluorescent quantitative analysis (for example, just a few). GAPDH is sometimes used as an internal control, but it is necessary to ensure that processing factors will not affect GAPDH. GAPDH, as a gene in the carbohydrate metabolism link, may be affected by toxins and will change expression, which has certain restrictions. So this the study only selected β-actin as the first choice internal control.
R1: The introduction lacks scientific soundness and important information e.g., what type of molecule is PD. Rationale must be improved (see question 1).
A:Thanks for the reviewer's question. PD is a monomer extracted from the dried rhizomes of traditional Chinese medicine Polygonum cuspidatum and belongs to the stilbene compound. It has the functions of relieving dampness and relieving yellow, clearing heat and detoxification, dispelling blood stasis and relieving pain, relieving cough and resolving phlegm. Modern pharmacology has proved that Polygonum cuspidatum has the functions of protecting the cardiovascular system, anti-oxidation and improving the body's immunity. Polydatin has the effect of protecting the myocardium and cardiovascular system, and can resist the damage of myocardial ischemia, thereby anti-shock. In addition, it has good effects in anti-pulmonary hypertension, lowering blood lipid, anti-thrombosis and protecting renal vascular damage. The pharmacological effects and various biological activities of polydatin have high research and application value. Regarding the detailed introduction of PD, we provide supplementary explanation in the preface. L69-74 for supplementary explanation. Polydatin (3, 4', 5-trihydroxystibene-3-β-mono-D-glucoside; PD) is a natural precursor of resveratrol, and is extracted from Polygonum cuspidatum for Chinese medicine [12]. PD is an effective ingredient of Chinese medicines such as Polygonum cuspidatum. It is the product of resveratrol and glucose combined, so it is also called resveratrol glycoside. Its pharmacological effects are similar to resveratrol and can be transformed into each other in the body. But PD has a stronger antioxidant effect and stability.
R2: Authors write (line 33-35): “Zearalenone induces apoptosis in 33 mouse Leydig cells by activating the endoplasmic reticulum (ER) stress-dependent signaling pathway [6].” However, I miss the connection among Laydig cells (that are in seminiferous tube and I important in spermatogenesis), mammary epithelial cells and milk yield and quality. Probably there is a connection, but authors must include this explanation in the introduction.
A:Thanks for the reviewer's question. Here we have revised the foreword.L44-50。“Studies have shown that ZEA induces mouse Leydig cell apoptosis by activating the endoplasmic reticulum (ER) stress-dependent signaling pathway. ZEA is detected in milk, and we think that ZEA may have similar effects on bovine mammary epithelial cells. The function of the ER is sensitive to unfolded protein accumulation, calcium homeostasis and redox state changes, and disruption of these processes can cause ER stress. Unfolded protein response (UPR) is an ER stress response that maintains cell homeostasis, and deregulation of UPR can cause disorders and cell death.”
R3: please, give rationale/explanation about usage of mammary epithelial cells and no other cells!
A:Thanks for the reviewer's question. First of all, the purpose of this study with bovine mammary epithelial cells is to see whether ZEA can interfere with polydatin when it has a toxic effect on bovine mammary epithelial cells. For this purpose, bovine mammary epithelial cells were chosen. We chose the MAC-T cells selected in this study. Primary culture and cell lines are useful in the study of bovine mammary epithelial cells. The primary culture will undergo self-apoptosis due to its own characteristics. If the work toxicity cannot be guaranteed because of the toxic effect or the cell itself. Therefore, this research requires the use of stable cell lines. MAC-T cells are a good bovine mammary epithelial cell model. Here are the advantages of MAC-T. MAC-T cells are a typical cell line for studying breast epithelial cells. They have multiple functions and are good at reflecting the lactation characteristics of breast cells. At present, many studies on breast inflammation and other studies are conducted with MAC-T. This research group also uses MAC-T cells to carry out research work such as AFB1, DON and other mycotoxins. References are listed below.
[1] Jing Zhang#, JunMei Wang#, HengTong Fang, Hao Yu, Yun Zhao, JingLin Shen, ChangHai Zhou, YongCheng Jin*. Pterostilbene inhibits deoxynivalenol-induced oxidative stress and inflammatory response in bovine mammary epithelial cells, Toxicon, 2021, 189:10–18
[2] YuRong Fu#, YongCheng Jin#, Yun Zhao, AnShan Shan, HengTong Fang, JingLin Shen, ChangHai Zhou, Hao Yu, YongFeng Zhou, Xin Wang, JunMei Wang, RuiHua Li, Rui Wang, and Jing Zhang*. Zearalenone induces apoptosis in bovine mammary epithelial cells by activating endoplasmic reticulum stress. Journal of Dairy Science. 2019, 102(11):10543–10553
[3]Junmei Wang#, Yongcheng Jin#, Shunlu Wu, Hao Yu, Yun Zhao, Hengtong Fang, Jinglin Shen, Changhai Zhou, Yurong Fu, Ruihua Li, Rui Wang, Junxiong Wang, Kexin Zheng, Qingsong Fan, Bojiong Chen, Jing Zhang*. Deoxynivalenol induces oxidative stress, inflammatory response and apoptosis in bovine mammary epithelial cells. Journal of Animal Physiology and Animal Nutrition. 2019, 103(6):1663–1674.
[4]Yongfeng Zhou#, Yongcheng Jin#, Hao Yu, Anshan Shan, Jinglin Shen, Changhai Zhou, Yun Zhao, Hengtong Fang, Xin Wang, Junmei Wang, Yurong Fu, Rui Wang, Ruihua Li, Jing Zhang*. Resveratrol inhibits aflatoxin B1-induced oxidative stress and apoptosis in bovine mammary epithelial cells and is involved the Nrf2 signaling pathway. Toxicon. 2019, 164: 10–15
R4: Introduction needs more detailed information on relationship between oxidative stress and endoplasmic reticulum (ET) stress.
A:Thanks for the reviewer's question. Regarding the detailed introduction of oxidative stress and endoplasmic reticulum stress., we provide supplementary explanations in the preface.L62-66.” The generation of ROS and oxidative stress can be considered as an indispensable part of UPR. Therefore, ROS generation can be both upstream and downstream of UPR. Both oxidative stress and ER stress are involved in a variety of physiological and pathophysiological conditions, and they play a vital role in cell homeostasis and apoptosis. This shows that there are many connections between ER stress and oxidative stress.”
R5: Each result undersection must be re-written giving exact numbers (where it is important) not only mentioning what is increased or decreased. Also start each section with short introduction on why authors look at certain parameters.
A:Thank you reviewers for their comments. We have fully revised the results section, please refer to the article for details.
R6: Choose only ONE TYPE of statistical significance. Usually, scientific graphs have the * symbol for p values.
A:Thanks for the reviewer's question. We have modified the significance of the result graph. Changed to "*" means.
R7: Line 72-73 “In Fig. 1C, they are the control group, 72 the ZEA treatment group, the PD treatment group, and the ZEA+PD treatment group.” – this statement belongs to materials and methods NOT in results.
A:The reviewers ask questions. We will make changes.
R8: Line 78 “LDH activity can serve as a proxy for cytotoxicity” – when mentioning for the first time, please, state the full name of LDH.
A:Thanks for the reviewer's question. We will make changes. L105, Lactate dehydrogenase activity.
R9: Line 81-83 “Compared with the ZEA-treated group, the LDH content in the cells treated with ZEA + PD was significantly reduced (Figure 2B, P<0.01).” – HOW much! Need numbers!
A:Thanks for the reviewer's question. We will make changes.L109-113” Compared with the control group, LDH increased by about 43% in the ZEA treatment group.Compared with the ZEA treatment group, the ZEA+PD treatment group decreased by about 23%.In MAC-T cells treated with ZEA, PD lessens the cytotoxic effect due to interactions of ZEA and PD.”
“PD reduces the LDH of MAC-T cells because ZEA increases.” – is this a conclusion? Why it is here. Also the sentence lacks scientific style. E.g. the results can lead to suggestion that in …cells treated with ZEA, PD lessens the cytotoxic effect due to interactions of ZEA and PD. -this is only an example.
A:Thanks for the reviewer's question. We will make changes. L112-113”.In MAC-T cells treated with ZEA, PD lessens the cytotoxic effect due to interactions of ZEA and PD.”
R10: Section 2.3. probable typo in the title, Cxidative stress – needs to be Oxydative stress. Also, what does MDA, T-SOD and GSH stands for and why they are important (probably it is better to mention it in introduction)?
A:Thanks for the reviewer's question. We will make changes. The introduction of MDA, T-SOD and GSH is added to the introduction. L38-43.” The balance between reduced glutathione (GSH) and oxidized glutathione (GSSH) is essential for maintaining redox homeostasis in cells. Due to the depletion of GSH, ROS is generated during protein misfolding. Parameters related to oxidative stress include: malondialdehyde (MDA), reduced glutathione (GSH) and total superoxide dismutase (T-SOD) with antioxidant enzyme activity and so on.”
R11: In method about measuring the levels of ROS, authors state that intensity was measured “using the fluorescence microscope function of a Cytation 5 cell imaging reader”. Does it include counting the cells? How many view fields were counted? Is it really quantitative or more semi0quantitative method? If it is quantitative methods, please provide actual (or average) numbers in each group including standard deviation!
A: The Cytation5 intelligent live cell imaging system is also a multifunctional microplate reader, which can be integrated, automated, and digital microscopic imaging. We widely choose random single-field ROS imaging. The instrument performs fluorescence analysis on the selected field and gives the corresponding fluorescence value. Intracellular ROS levels in MAC-T cells were measured by using DCFH-DA and a kit to measure active ROS. The level of ROS was demonstrated by the green fluorescence intensity of cells.
R12: Figure 4 need more visible scale bars! Also, in the legend for figure 4: “In the figure, A and B means P<0.01” – how can two signs mean the same p value?
A:Thanks for the reviewer's question. We adjust the scale of Figure 4A to be clearly visible. Regarding the saliency representation of Figure B, we have used "*" in the representation of all result images, which is specifically reflected in the article.
R13: Section about ET stress and gene expression – please provide (either in introduction or in discussion or even here) the information-rationale to use those particular genes as ET stress markers. Also, mention fold changes!
A:Thanks for the reviewer's question. Regarding the issue of gene selection for endoplasmic reticulum stress, we will make a supplementary explanation in the introduction. L50-61.” However, through previous studies, it was found that ZEA would indeed induce apoptosis of MAC-T cells. Mammalian cells express the UPR transducers IRE1, PERK, and ATF6, which control transcriptional and translational responses to ER stress. Under non-stress or physiological conditions, these proteins remain in an inactive state and bind to the molecular chaperone BiP/GRP78, which is also known as the main regulator of ER stress. Under ER stress, GRP78 decomposes from the stress sensors, causing them to activate. Once activated, PERK will cause the phosphorylation of the α subunit of the translation initiation factor eIF2α (eukaryotic initiation factor 2α), which leads to the overall reduction of translation and allows the UPR-dependent genes (such as activated transcription factor 4 (ATF4)) Priority translation. The important target of ATF4 is CHOP (C/EBP homologous protein).
”
R14: Figure 5 – for better visibility all graphs should have identical Y axis scale. Consistency!
A: Thanks for the reviewer's question. We will uniformly change the Y-axis scale of all graphs in Figure 5.
R15: Primer table must be in section Materials and Methods.
A: Thanks for the reviewer's question. We moved the table to materials and methods.
R16: Section 2.6. about apoptosis: authors should start with explanation how the apoptosis was detected and then continue with findings. About Bax and Bcl2 – at least on one place mention which one was PRO and which was ANTI apoptotic protein. Please, remember that gene names must be written in italica.
A: Thanks for the reviewer's question. We further explain the apoptosis genes and change the names of all genes to italics. L269-279.“The CCAAT/enhancer binding protein homolog protein (CHOP) is a key pro-apoptotic transcription factor related to ER stress. As we all know, Bcl-2 gene is an anti-apoptotic gene, and Bax gene is a pro-apoptotic gene. CHOP-mediated ER stress inhibits Bcl-2 expression, promoting expression of Bax to induce apoptosis. We found that cells exposed to ZEA had lower Bcl-2 expression level and elevated Bax expression level. However, adding PD increased Bcl-2 expression and decreased Bax expression. Flow cytometry mediated measurement of apoptotic rate showed that ZEA treatment significantly increased apoptosis compared to the control group, however, addition of PD significantly reduced the apoptosis rate. Although PD significantly reduced the rate of cell apoptosis, the degree of reduction was not large, indicating that 5 μM PD can reduce the occurrence of cell apoptosis, but it cannot completely inhibit it.”
R17: Figure 6 A: at least in legend, provide information what is what in those FACS dot plots? Also, each quadrant description (e.g. in control dot plot Q1-UL, Q1-UR etc is very confusing. Name them correctly. In addition, I would suggest to put pictures A and B beside each other and then all three graphs with gene expression changes and caspase activity in other line. Probably it is a fold change depicted in figure 6. C and D – then it must be mentioned in the results and also number must be given with standard deviations.
A: Thanks for the reviewer's suggestion. We accept revisions suggested by reviewers. L207-212“Figure 6. Effect of PD on ZEA-induced apoptosis in MAC-T cells. (a) Apoptosis measurements using annexin V / PI. The lower left panel contains annexin V and PI negative cells, the lower right panel is annexin V positive. The upper left panel is annexin V- and PI+, and the upper right panel is annexin V+ and PI+ (b)Four groups of apoptotic rate. (c-d) q-PCR analysis of mRNA expression levels of apoptosis-related genes Bcl-2 and Bax. (e) Four groups of Caspase-3 activity. In the figure, “*” means P<0.05. “**”means P<0.01. “***”means P<0.001.L185-188.“We found that treating MAC-T cells with ZEA increased the apoptotic rate of cells compared to the control group (P<0.001, Figures 6A-B). Increased from 10.22% to 29.67%.ZEA + PD treatment group had significantly lower apoptotic rate than the ZEA treatment group (P<0.01). Reduced from 29.67% to 29.06%.”
R18: Discussion beginning (line 155): If it (mycotoxin) is in the milk, how it gets there? And is it still harmful for us as milk-drinkers? Or is the problem here only about cows giving more milk?
A: Thanks for the reviewer's question. We will make additional explanations in the discussion section. L217-227.“From the current stage of the dairy production process, governments of all countries have clear detection limits for toxins in milk, which cannot be higher than this detection value. So normal milk is not harmful. But as a dairy cow, will toxins be harmful to the dairy cow itself when toxins enter the milk through the mammary gland? From the perspective of animal welfare, it is best to reduce the content of toxins in feed to ensure that cows are not affected by toxins. However, in actual production, certain toxins may enter the mammary glands and cause toxic effects on mammary cells. Therefore, it is necessary to carry out research on the direct toxicity of ZEA on breast cells from the cellular level, and to seek the protection of plant compounds to protect breast cells from such oxidative stress, which affects milk production performance.”
R19: line 163 - Again, what was the rationale to choose PD? Please provide the information. What type of molecule is PD?
A: Thanks for the reviewer's question. PD is a monomer extracted from the dried rhizomes of traditional Chinese medicine Polygonum cuspidatum and belongs to the stilbene compound. It has the functions of relieving dampness and relieving yellow, clearing heat and detoxification, dispelling blood stasis and relieving pain, relieving cough and resolving phlegm. Modern pharmacology has proved that Polygonum cuspidatum has the functions of protecting the cardiovascular system, anti-oxidation and improving the body's immunity. Polydatin has the effect of protecting the myocardium and cardiovascular system, and can resist the damage of myocardial ischemia, thereby anti-shock. In addition, it has good effects in anti-pulmonary hypertension, lowering blood lipid, anti-thrombosis and protecting renal vascular damage. The pharmacological effects and various biological activities of polydatin have high research and application value. Regarding the detailed introduction of PD, we provide supplementary explanation in the preface. L69-74 for supplementary explanation. Polydatin (3, 4', 5-trihydroxystibene-3-β-mono-D-glucoside; PD) is a natural precursor of resveratrol, and is extracted from Polygonum cuspidatum for Chinese medicine [12]. PD is an effective ingredient of Chinese medicines such as Polygonum cuspidatum. It is the product of resveratrol and glucose combined, so it is also called resveratrol glycoside. Its pharmacological effects are similar to resveratrol and can be transformed into each other in the body. But PD has a stronger antioxidant effect and stability.
R20: line 176 “… to treat health-related diseases” – do we have health-non related diseases? Line 177 “…kill various human tumor cell lines” – styling is poor.
A: Thanks for the comments of the reviewers, we correct this sentence. L244-246.It has been reported that PD can improve the damage of arsenic to rat testicular cells, and PD can also inhibit the growth of some human tumor cells.
R21: Discussion has a lot of repetitious information e.g. lines 178-180 (about LDH) repeats Lines 168-170.
A:Thanks for the reviewer's question. We will delete duplicate parts.
R22: Line 187-189 “The effect of ZEA on GPx, CAT, and SOD in the 187 testis tissue of adult Balb/c male mice was significantly decreased, while MDA in the 188 same tissue was significantly increased [27].” – How would you explain that increase?
A: Thanks for the reviewer's question. MDA is malondialdehyde. Under normal circumstances, the expression of MDA will not increase, and the expression of Gpx, CAT and SOD will not decrease. But when oxidative stress occurs, MDA will increase and antioxidant enzyme activity will decrease. GSH and T-SOD will decrease. Then our study showed that ZEA increased MDA, decreased GSH and T-SOD, etc., indicating that oxidative stress occurred in breast cells. The protection of PD reduces oxidative stress damage.
R23: Lines 190-192 “Exposing cells to both ZEA and PD reduced MDA content and increased GSH and T-SOD content [28, 24].” – but is it done in rats! Please be precise and mention full information. Also, the next sentence states the similarities of your work with Jiang et al. [15]. Please, provide in which model they observed the statement. How it related with previous sentence?
A: Thanks for the reviewer's question. We add an explanation in the discussion section. L258-263“However, even the lowest concentrations of PD that we tested alleviated the oxidative damage caused by ZEA on MAC-T cells. Exposing cells to both ZEA and PD reduced MDA content and increased GSH and T-SOD content. Studies have shown that PD can reduce oxidative damage in cardiomyocytes. This result is similar to ours. PD reduces the oxidative damage of MAC-T cells caused by ZEA. ”
R24: while giving some reference on GRP78 and CHOP as possible ET stress markers, nothing is said about ATF4, ATF6 and ASK1.
A: Thanks for the reviewer's question. Regarding the issue of gene selection for endoplasmic reticulum stress, we will make a supplementary explanation in the introduction. L50-61.” However, through previous studies, it was found that ZEA would indeed induce apoptosis of MAC-T cells. Mammalian cells express the UPR transducers IRE1, PERK, and ATF6, which control transcriptional and translational responses to ER stress. Under non-stress or physiological conditions, these proteins remain in an inactive state and bind to the molecular chaperone BiP/GRP78, which is also known as the main regulator of ER stress. Under ER stress, GRP78 decomposes from the stress sensors, causing them to activate. Once activated, PERK will cause the phosphorylation of the α subunit of the translation initiation factor eIF2α (eukaryotic initiation factor 2α), which leads to the overall reduction of translation and allows the UPR-dependent genes (such as activated transcription factor 4 (ATF4)) Priority translation. The important target of ATF4 is CHOP (C/EBP homologous protein).
R25: Materials and Methods – Line 265: “The cell fluid was collected….” – did you mean “cell media” or “cell supernatant”?
A: Thanks for the reviewer's question. We are using cell supernatant, which is explained in the article.
R26: Section 4.5. Detection of GSH, T-SOD and MDA levels – this section need more explanation. I will not go and look for specific protocol (manufacturer must be mentioned in methods, not only in materials).
A: Thanks to the reviewers for their questions, and we will add additional explanations about the methods of detecting GSH, T-SOD and MDA.L346-361.“The method to determine MDA.After the cells have been cultured, collect the culture solution for later use. The cells were taken out with a cell scraper and transferred to a 1.5 mL centrifuge tube. Add 500μL of the extract, use a glass homogenizer to make a mixture, take 100ul and transfer to another 1.5 mL centrifuge tube. The BCA kit determines the protein concentration. Take the 96-well plate and set the blank, standard and measurement tubes, and add reagents according to the instructions. Measure the absorbance at 530nm in the microplate reader and record the data. Method of measuring T-SOD. Cultivate the cells in the same way, and measure the cell protein concentration after finishing. Use the kit instructions to add the reagents, mix well and let stand at room temperature for 10 min, colorimetrically detect the wavelength of 550nm, and make a record. Method for determining GSH. After the culture, the cells were taken out by cell scraping and transferred to a 1.5 mL centrifuge tube. Use a glass homogenizer for mixing. Take 100μL of the precipitation solution, centrifuge at 3500r/min for 10min, and take the supernatant for detection. Use a 96-well plate, add reagents according to the instructions, measure the absorbance at 405nm in a microplate reader, and make a record. ”
R27: Section 4.6. is very confusing: Line 277-279 “When the cells confluence [strange styling] 70-80 %, they were treated and cultured in a medium containing ZEA or PD for 24 h. Then they were divided into control, ZEA, PD and ZEA+PD treatment groups.” – Grouping would be logical before treatment, but how were you able to take control group, if you treated cells with ZEA or PD?
A:Thanks for the reviewer's question. We will make changes. Here to explain, Randomly divide into three groups, each with 3 repeats. The experiments were divided into Control, ZEA, PD and ZEA + PD treatment groups.
R28: Section 4.7. qPCR - How much RNA was used for cDNA synthesis? Why two RT reagents? Why repeat what was written in section 4.1.? Do not write it in 4.1. section then. Line 289 “First strand cDNA was synthesized according to manufacturer’s instructions.” - which manufacturer? In my opinion this sentence is irrelevant.
A: Thanks for the reviewer's question. We will make changes.L373-375“Total RNA was isolated using TRIzol reagent (Invitrogen, Carlsbad, CA). Use 1μg of RNA to synthesize cDNA at 42°C for 15 min, and then at 85°C for 5 min. SYBR Green Mix Kit is used to perform q-PCR reactions.”
R29: please, provide the reference also for the Fold change method (delta delta Ct).
A: Thanks for the reviewer's question. For the q-PCR calculation method, we use 2-△△Ct.
R30: Section 4.8. and 4.9. please be consistent with description of the speed of centrifugation, using either “rpm” or more widely used “… x g”.
A:Thanks for the reviewer's question. 4.8 is the detection of apoptosis by flow cytometry. According to the kit instructions, the relevant centrifugation instructions are marked as "rpm". However, 4.9 is a Caspase-3 activity test, and the kit instructions are marked as "...x g". Because there are different kits and different test methods, the unit of centrifugal speed is not uniform.
R31: Conclusions are to general and needs to be more conclusive (for that you need a clear rationale and need to give bigger picture (see Q1)).
A:Thanks for the reviewer's question. We rewrite the conclusion part.L413-421.“In conclusion, in vitro experiments revealed a new phenomenon that PD reduces oxidative damage and ER stress in MAC-T cells induced by ZEA by reducing the pro-duction of ROS, the activity of antioxidant enzymes and the expression of ER stress -related genes. The occurrence of stress finally alleviates cell apoptosis. These findings indicate that PD can be used as an effective antioxidant and potential therapeutic agent for diseases related to oxidative stress and ER stress. However, the potential molecular mechanism of PD's protective effect on ZEA-induced cell damage in MAC-T cells needs further research. In vivo studies are needed to determine the role of PD as an effective therapeutic agent, which can be used as a feed additive.”
Round 2
Reviewer 2 Report
Comments and Suggestions for Authors
The manuscript (ID:toxins-1084421) “Polydatin protects bovine mammary epithelial cells upon Zearalenone-induced apoptosis through inhibiting oxidative responses and endoplasmic reticulum stress” has been revised and partially corrected. However, while the authors gave satisfactory answers to some questions related to scientific part of this manuscript, and scientific content is satisfactory, main major concern still is the styling and language issues.
Here are some comments and one question for authors:
Q1: While Authors show the dose-dependent (on some level) ZEA effect on cell viability, why did they choose 30 mkM concentration for further experiments, but no 120 mkM that shows much more profound effect on cell viability? What would be the expected ZEA concentration range in actual cow feed?
Introduction could contain the future perspectives of this research (as the author’s answered on my 1st question).
I do not want to write every single revision, however, already in section “Key Contribution” the sentence: “The toxic damage of ZEA to bovine mammary epithelial cells.” Probably would be better as: “ZEA causes the toxic damage to bovine mammary epithelial cells, leading to various effects….”
Just as an example:
Introduction.
Sentence: “Zearalenone is cytotoxic [1], and toxic effects on the liver and spleen [2], intestines [3] and hematopoietic cells [4].” does not sound as correct scientific language styling. Perhaps “Zearalenone is cytotoxic [1], and its toxic effects have been described in the liver, spleen [2], intestines [3] and hematopoietic cells [4].
Sentence: “ZEA is detected in milk, and we think that ZEA may have similar effects on bovine mammary epithelial cells.” – while after the author’s answers to my questions I kind of understand what they mean here, however, it might be a bizarre sentence for reader.
Introduction still as not flowing like a story.
Results
2.1. PD Increases the Activity of MAC-T Cells Due to ZEA - the title should be changed, if you talk only about viability!
I still want to see the numbers at least in the text, for example for Figure 1 C. While I can see that there is some decrease in cell viability in cells cultured in presence of ZEA (100% in control and something like 85% in ZEA), it is hard to understand the viability of the cells cultured in presence of both ZEA and PD. It seems it is still not back to 100%.
2.2. PD Reduces the Activity of LDH in ZEA-Induced MAC-T cells – I am not the author, but I would give the title clearer, like “PD reduces the ZEA-induced cytotoxic effect in MAC-T cells”.
2.3. section
“PD increased the decrease in cell viability caused by ZEA and decreased the increase in LDH caused by ZEA. Therefore, we have noticed that it may cause oxidative damage.”
– while I understand what authors want to say here, the sentences must bet revised.
For Figure 6 a – it would be easier to understand the panels if on Y and X axis the PI and Annexin V would be instead of PE and FITC.
Discussion
Sentences: “So normal milk is not harmful. But as a dairy cow, will toxins be harmful to the dairy cow itself when toxins enter the milk through the mammary gland?” should be rewritten. I do not understand the second sentence at all. Perhaps they can be left out?
Rationale is given (Line 583-Line 86), but the way it is written, is unsatisfactory.
Materials and methods
4.3. Cell viability assay – what do the authors mean by “Randomly divide into three groups, each with 3 repeats” – we see in following sentence that there are 4 groups (Control, ZEA, PD and ZEA+PD).
4.4. Lactate Dehydrogenase Assay
Line 840-Line 841 – Authors write that cell culture media was collected for viability test, but isn’t the LDH assay the assay to test cytotoxicity in this particular study?
4.5. Detection of GSH, T-SOD and MDA levels – in previous review it was asked to describe methods in brief – meaning, that no need for full protocols as it is now. It would be enough to describe the principle etc.
Author Response
Dear reviewer,
Thank you for your many good suggestions for our manuscript. We accept your proposal to better improve our manuscript. Our article has been professionally optimized for language, and a proof is attached. Hope to help you and readers to read our article.
Below we will answer your questions.
Q1: While Authors show the dose-dependent (on some level) ZEA effect on cell viability, why did they choose 30 mkM concentration for further experiments, but no 120 mkM that shows much more profound effect on cell viability? What would be the expected ZEA concentration range in actual cow feed?
A: Thanks for the reviewer's question. First, we treated MAC-T cells with different concentrations of ZEA for 24 h. The cell viability results showed that compared with the control group, the 30 μM ZEA group was extremely significantly reduced (P <0.001). The results of lactate dehydrogenase detection of cytotoxicity showed that compared with the control group, the 30 μM ZEA group was significantly higher (P <0.01). Whether it is cell viability testing or lactate dehydrogenase testing, 30uM ZEA is the most obvious concentration.
Secondly, in actual production, ZEA can be detected in milk. According to the referenced literature, the concentration of ZEA detected in milk ranges from 0.51 to 5.09 μg/L. According to the basic theory of milk synthesis in the breast, we know that it takes about 400 liters of blood to produce 1 liter of milk. Calculated based on the highest ZEA concentration detected in milk of 5.09μg/L, this means that the ZEA content in 400L of blood should be at least 5.09μg before it can be transferred to milk. Then we can push back the actual concentration of ZEA in the blood: 5.09μg÷400L = 0.017225μg/L, the molecular weight of ZEA is 318.36g/mol, then the molar concentration of ZEA in the blood is: 0.017225μg/L ÷318.36 g/mol = 39.97μM. It shows that the 30μM concentration selected by the cell assay has physiological significance within the actual biological dose range.
Finally, according to an article "Occurrence, Impact on Agriculture, Human Health, and Management Strategies of Zearalenone in Food and Feed: A Review" in Toxins magazine, the detected content of ZEA in milk is 1.0–11.9 µg/kg (Dipendra Kumar Mahato et al.2021). We choose The 30 μM ZEA is fully in line with this range. Check out a large number of relevant ZEA in vitro research experiments on cells, and the concentration we chose is included in this range. Then, according to the calculation of the concentration in the milk, and considering the monitoring concentration range in the feed, as close as possible to the actual situation, we are considering the protection research of the toxic effect of the low density range (within the actual physiological concentration) on the mammary gland.
Table 1.
|
TM4 cells |
1-100 μM |
(Zheng et al., 2018) |
|
Caco-2、DOK and Vero cells |
10-40 μM |
(Abid et al., 2003) |
|
Sertoli cells |
15-60 μM |
(Zheng et al., 2016) |
References
Dipendra Kumar Mahato, Sheetal Devi, Shikha Pandhi 3 , Bharti Sharma , Kamlesh Kumar Maurya ,Sadhna Mishra, Kajal Dhawan , Raman Selvakumar , Madhu Kamle, Awdhesh Kumar Mishra and Pradeep Kumar. Occurrence, Impact on Agriculture, Human Health, and Management Strategies of Zearalenone in Food and Feed: A Review. Toxins 2021, 13, 92. https://doi.org/10.3390/toxins13020092
Abid, E. S., I. W. Baudrimont, Z. Ouanes, T. A. Mobio, R. Anane, E. E. Creppy, and H. Bacha. 2003. DNA fragmentation, apoptosis and cell cycle arrest induced by zearalenone in cultured DOK, Vero and Caco-2 cells: prevention by Vitamin E. Toxicology 192(2):237-248.
Zheng, W., S. Pan, G. Wang, Y. J. Wang, Q. Liu, J. Gu, Y. Yuan, X. Z. Liu, Z. P. Liu, and J. C. Bian. 2016. Zearalenone impairs the male reproductive system functions via inducing structural and functional alterations of sertoli cells. Environmental toxicology and pharmacology 42:146-155.
Zheng, W. L., B. J. Wang, L. Wang, Y. P. Shan, H. Zou, R. L. Song, T. Wang, J. H. Gu, Y. Yuan, and X. Z. Liu. 2018. ROS-Mediated Cell Cycle Arrest and Apoptosis Induced by Zearalenone in Mouse Sertoli Cells via ER Stress and the ATP/AMPK Pathway. Toxins 10(1):24.
I do not want to write every single revision, however, already in section “Key Contribution” the sentence: “The toxic damage of ZEA to bovine mammary epithelial cells.” Probably would be better as: “ZEA causes the toxic damage to bovine mammary epithelial cells, leading to various effects….”
A: Thanks to the reviewers’ comments, we have revised the Key Contribution. L31-33.“ZEA induces oxidative damage to bovine mammary epithelial cells, leading to various effects. PD can alleviate the oxidative stress response of bovine mammary epithelial cells to ZEA by reducing the subsequent ER stress response.”
Introduction.
Sentence: “Zearalenone is cytotoxic [1], and toxic effects on the liver and spleen [2], intestines [3] and hematopoietic cells [4].” does not sound as correct scientific language styling. Perhaps “Zearalenone is cytotoxic [1], and its toxic effects have been described in the liver, spleen [2], intestines [3] and hematopoietic cells [4].
A: Thanks to the reviewer's comments, we have made changes. L38-38.“Zearalenone is cytotoxic [1] to the liver, spleen [2], intestines [3], and hematopoietic cells [4].”
Sentence: “ZEA is detected in milk, and we think that ZEA may have similar effects on bovine mammary epithelial cells.” – while after the author’s answers to my questions I kind of understand what they mean here, however, it might be a bizarre sentence for reader.
A: Thanks to the reviewer's comments, we have made changes.L54-55.“ZEA was previously detected in milk samples. Therefore, whether ZEA may have similar effects on bovine mammary epithelial cells is worth studying.”
Results
2.1. PD Increases the Activity of MAC-T Cells Due to ZEA - the title should be changed, if you talk only about viability!
I still want to see the numbers at least in the text, for example for Figure 1 C. While I can see that there is some decrease in cell viability in cells cultured in presence of ZEA (100% in control and something like 85% in ZEA), it is hard to understand the viability of the cells cultured in presence of both ZEA and PD. It seems it is still not back to 100%.
A: Thanks to the reviewers’ comments, we have revised the title of 2.1.
L93.“2.1. The increase in PD decreased the MAC-T cell viability due to ZEA.”
The content we add to this part is “The cell viability of the ZEA treatment group decreased by about 13%, relative to the control group.Compared with the ZEA treatment group, the ZEA+PD treatment group increased by about 9%.The addition of PD effectively increased cell viability due to ZEA inhibition”。
Regarding the explanation of cell power, we have added PD to improve cell viability, but it does not mean that PD has a 100% ability to improve, but compared to ZEA group, PD+ZEA group can increase %. 100% effect is unscientific.
2.2. PD Reduces the Activity of LDH in ZEA-Induced MAC-T cells – I am not the author, but I would give the title clearer, like “PD reduces the ZEA-induced cytotoxic effect in MAC-T cells”.
A:Thanks to the reviewers’ comments, we have revised the title of 2.2. L114. “2.2. PD reduces the ZEA-induced cytotoxic effect in MAC-T cells”
2.3. section
“PD increased the decrease in cell viability caused by ZEA and decreased the increase in LDH caused by ZEA. Therefore, we have noticed that it may cause oxidative damage.”
– while I understand what authors want to say here, the sentences must bet revised.
A: Thanks to the reviewers for their comments, we have revised this part of the content. L148-149. “These results indicate that PD can inhibit the oxidative damage caused by ZEA in MAC-T cells.”
For Figure 6 a – it would be easier to understand the panels if on Y and X axis the PI and Annexin V would be instead of PE and FITC.
A: Thank you reviewers for their comments. First of all, PI and Annexin V are staining methods, or double staining methods, which use labeled fluorescence to identify cell apoptosis. However, PE and FITC are detection channels. Therefore, it is reasonable to show as shown in Figure 6a.
Discussion
Sentences: “So normal milk is not harmful. But as a dairy cow, will toxins be harmful to the dairy cow itself when toxins enter the milk through the mammary gland?” should be rewritten. I do not understand the second sentence at all. Perhaps they can be left out?
Rationale is given (Line 583-Line 86), but the way it is written, is unsatisfactory.
A: Thanks to the reviewers for their comments, we have revised this part. L229-236. “Currently, most countries have clear guidelines and detection limits for milk toxins. Although ZEA was detected in milk, the content may not be harmful to the human body. But for the dairy cow itself, after feeding the feed containing ZEA, if the ZEA is transferred to the milk through the bovine mammary gland, it cannot be ignored that ZEA is harmful to the dairy cow itself. Therefore, it is necessary to evaluate the toxic effects of ZEA on breast cells at the cellular level, and seek ways to protect breast cells from this damage, which is beneficial to the subsequent impact on milk production.”
Materials and methods
4.3. Cell viability assay – what do the authors mean by “Randomly divide into three groups, each with 3 repeats” – we see in following sentence that there are 4 groups (Control, ZEA, PD and ZEA+PD).
A: Thanks to the reviewers for their comments, we have revised this part. L333-335. “Randomly divide into four groups, each with three repeats. The experiments included control, ZEA, PD and ZEA + PD treatment groups.”
4.4. Lactate Dehydrogenase Assay
Line 840-Line 841 – Authors write that cell culture media was collected for viability test, but isn’t the LDH assay the assay to test cytotoxicity in this particular study?
A:Thanks to the reviewers for their comments, we have revised this part. L341-342.“the cell culture supernatant was collected for cytotoxicity level detection. “
4.5. Detection of GSH, T-SOD and MDA levels – in previous review it was asked to describe methods in brief – meaning, that no need for full protocols as it is now. It would be enough to describe the principle etc.
A: Thanks to the reviewers for their questions, we made changes to GSH, T-SOD and MDA. L346-360.“The levels of GSH, T-SOD, and MDA were tested using the respective assay kits, and following the manufacturer’s protocol. As described above, MAC-T cells were treated with varying concentrations of ZEA and PD for 24 h and cells were collected to measure oxidation levels. The culture solution was also collected for later use. The cells were taken out with a cell scraper and transferred to a 1.5 mL centrifuge tube. Then, 500 μL of the extract was added and the contents mixed by homogenization. Subsequently, 100 μL of the mixture were transferred to another 1.5 mL centrifuge tube. The BCA kit determines the protein concentration. Measure the absorbance at 530 nm in the microplate reader. To determine the T-SOD content, the cells were cultivated in a similar way, and then the cell protein concentration was evaluated. Use the kit instructions to add the reagents, mix well and let stand at room temperature for 10 min, colorimetrically detect the wavelength of 550 nm. To determine the GSH content, cultured cells were taken out by cell scraping and transferred to a 1.5 mL centrifuge tube. Use a glass homogenizer for mixing. Take 100 μL of the precipitation solution, centrifuge at 3500 rpm for 10 min, and take the supernatant for detection. Measure the absorbance at 405 nm in the microplate reader. ”
Submission Date
05 January 2021
Date of this review
27 Jan 2021 14:52:53

Round 3
